# Differential Screening of Herniated Lumbar Discs Based on Bag of Visual Words Image Classification Using Digital Infrared Thermographic Images

**DOI:** 10.3390/healthcare10061094

**Published:** 2022-06-13

**Authors:** Gi Nam Kim, Ho Yeol Zhang, Yong Eun Cho, Seung Jun Ryu

**Affiliations:** 1Department of Spinal Neurosurgery, Gangnam Severance Hospital, Yonsei University College of Medicine, Seoul 06273, Korea; gnkim21@yuhs.ac (G.N.K.); yecho@yuhs.ac (Y.E.C.); 2Department of Neurosurgery, National Health Insurance Service Ilsan Hospital, Yonsei University College of Medicine, Goyang 10444, Korea; hyzhang@nhimc.or.kr

**Keywords:** bag of visual words, infrared thermography, lumbosacral radiculopathy, machine learning

## Abstract

Doctors in primary hospitals can obtain the impression of lumbosacral radiculopathy with a physical exam and need to acquire medical images, such as an expensive MRI, for diagnosis. Then, doctors will perform a foraminal root block to the target root for pain control. However, there was insufficient screening medical image examination for precise L5 and S1 lumbosacral radiculopathy, which is most prevalent in the clinical field. Therefore, to perform differential screening of L5 and S1 lumbosacral radiculopathy, the authors applied digital infrared thermographic images (DITI) to the machine learning (ML) algorithm, which is the bag of visual words method. DITI dataset included data from the healthy population and radiculopathy patients with herniated lumbar discs (HLDs) L4/5 and L5/S1. A total of 842 patients were enrolled and the dataset was split into a 7:3 ratio as the training algorithm and test dataset to evaluate model performance. The average accuracy was 0.72 and 0.67, the average precision was 0.71 and 0.77, the average recall was 0.69 and 0.74, and the F1 score was 0.70 and 0.75 for the training and test datasets. Application of the bag of visual words algorithm to DITI classification will aid in the differential screening of lumbosacral radiculopathy and increase the therapeutic effect of primary pain interventions with economical cost.

## 1. Introduction

In general, tests to diagnose lumbosacral radiculopathy include physical examination, magnetic resonance imaging (MRI), and electromyography. However, radiculopathy cannot be excluded from normal physical examination [1], and the diagnostic value of each of these findings is insufficient [2]. MRI is more expensive than computed tomography (CT), some researchers have insufficient MRI findings to diagnose radiculopathy [3], and many asymptomatic subjects are misdiagnosed with herniated discs based on MRI findings [4]. Electromyography is a useful method to diagnose radiculopathy [5]. However, there is an invasive aspect, which involves inserting a needle into the patient, thereby causing pain during the examination. Otherwise, digital infrared thermographic imaging (DITI) showed the functional changes of body temperatures noninvasively with economical cost and short examination time.

In lumbar herniated discs, hypothermia is induced in the lesion area when compared with the normal tissue because of the dysfunction of the autonomic nervous system [6]. DITI can objectively visualize changes in skin temperature in patients [7,8], especially those undergoing radiculopathy [9]. DITI provides an accurate diagnosis by using a thermodermatome to identify nerve roots; this helps clinicians in determining a treatment plan for lumbar discs [10]. Even though it uses thermatological characteristics, general practitioners and nurses find it difficult to accurately read a DITI.

Research on artificial intelligence and deep learning is being conducted to improve accuracy or to help diagnosis and treatment in various fields of medicine (radiology [11], ultrasound [12], pathology [13], and others). In anesthesia, research is being conducted to provide stable anesthesia through artificial intelligence and machine learning [14]. Artificial intelligence supports the development of predictive models to estimate the risk of diabetes and related complications, and it will help bring an element of personalized care in diabetes management [15]. Recent studies have shown that deep learning becomes a very promising adjunct in liver imaging tasks, which improves performance in detecting and evaluating liver lesions, facilitating clinical treatment for liver ailments, and predicting liver treatment response [16].

This study aimed to perform differential screening of L4-5 or L5-S1 discs, which are high-frequency herniated lumbar discs (HLDs) [17,18], and patients from two to five decades have a more than 90 percent chance of HLD occurring either at L4/5 or L5/S1 radiculopathy [19]. Because the dermatome is adjacent and it can be difficult to distinguish it, the authors try to perform the analysis by applying machine learning such that community medical personnel who are not spine specialists can help in conservative care, such as foraminal block, for patients who were classified based on DITI without using expensive equipment, such as MRI. Therefore, this machine learning algorithm helps clinicians accurately determine the target of foraminal root block with the screening of HLD level. The authors explored the infrared thermographic images of lumbosacral radiculopathy from a multicenter database using machine-learning-based infrared thermographic software.

## 2. Materials and Methods

### 2.1. Dataset and IRBs

A total of 1282 patients who performed one-level discectomy from 2006 to 2020 were retrospectively reviewed by specially trained spine surgeons, and 842 patients were enrolled in this study. The use of patient data retrospectively for research purposes was approved, and informed consent was waived by the Institutional Review Board at the National Health Insurance Service Ilsan Hospital (IRB no.: 2021-07-030). The dataset (National Reference Standard Korean Thermal Data Center at the National Health Insurance Service Ilsan Hospital and Gangnam Severance Hospital) of patients who underwent one-level discectomy was applied to develop and to externally validate a machine-learning-based classification tool for thermographic images.

### 2.2. Inclusion and Exclusion Criteria

The inclusion criteria were: (1) patients who were diagnosed with one of the various lumbar diseases, including lumbar disc herniation, at Ilsan Hospital from 2 March 2000 to 30 June 2021, and underwent DITI of the lower extremity; (2) those with at least one of the following radiologic criteria preoperatively: diagnosed with HLD based on CT or MRI images; and (3) those who findings were cross verified by a radiology specialist. The included patients were divided into two groups according to the presence or absence of HLD and whether or not they were classified as multiclass L4, L5, S1, or not.

The exclusion criteria were: (1) patients with diseases that can affect the skin temperature of the lower extremities, such as diabetes, peripheral arterial occlusive disease, and other endocrinology diseases; (2) those who underwent previous cervical or lumbar surgery; and (3) those who had HLD L4/5 or L5/S1 with massive downward or upward migration with different-level radiculopathy.

### 2.3. Clinical Data Collection, Labeling, Preprocessing, and Bag of Words Machine-Learning-Based Classification Modeling

Each center retrospectively collected the clinical data, including information on diagnosis using DITI. DITI included lower extremity sequences, such as leg anterior, posterior, and feet. The authors labeled the data HLD L4/5 patients undergoing DITI anterior, feet, and posterior as 1, 2, and 3 sequentially, respectively, and HLD L5/S1 patients undergoing DITI anterior, feet, and posterior were labeled 4, 5, and 6, respectively (Figure 1). The authors split the DITI dataset into a 7:3 ratio as the training algorithm and test dataset to evaluate model performance. In general, before training using machine learning, a pre-processing process that makes an image required for the input process is mandatory, and it is a very important process because the result of the output data after machine learning varies depending on the data used for modeling. In this study, the DITI data were standardized by removing the mean and scaling to unit variance with a python toolbox (sklearn. preprocessing. StandardScaler). For applying the bag of words technique for image classification, Matlab (The Math Works, Inc., MATLAB, version R2022a, Natick, MA, USA) Computer Vision Toolbox™ functions for image category classification by creating a bag of visual words were used. This process produces a histogram of visual word occurrences representing the image. These histograms are applied to train a labeled image classifier. The steps are feature extraction, which selects feature point locations using the grid method, and extracting speeded up robust features from the selected feature point locations, k-means clustering (initial guess of 6,928,800 points and 500 centroid), codebook generation, and learning and recognition (making the feature vector and support vector machine (radial basis function kernel) classify an image set) [20]. As a computer environment for machine learning, an algorithm for classifying lumbar disc herniation at L4/5 and L5/S1 was evaluated using Python and Matlab. The central processing unit was AMD Ryzen 7 PRO 4750G, the graphics processing unit (GPU) was NVIDIA’s Geforce RTX 3090 24 GB, and 64 GB of RAM was used.

## 3. Results

A total of 842 patients participated in this study; 534 patients were categorized as HLD L4/5 and 308 as L5/S1. Patient characteristics, including sex and age, are shown in Table 1. A slightly higher proportion of patients were aged over 50 years. Male patients were more predominant than female patients among those aged under 50 years, and female patients were more predominant than male patients among those aged over 50 years.

For differential screening of HLD L4/5 and HLD L5/S1, the bag of visual words algorithm was applied to DITI. Extracting 6,928,800 features from DITI kept 80% of the strongest features from each category. Image category 6 has the least number of strongest features (*n* = 847,024). Using the strongest features from each of the other image categories, k-means clustering was performed to create a 500-word visual vocabulary (Figure 2).

Training an image category classifier for six categories and the model performance of the machine learning algorithm were evaluated as a confusion matrix of the training and test sets. Model performance for HLD L4/5 patients undergoing DITI anterior, feet, and posterior, and for HLD L5/S1 patients undergoing DITI anterior, feet, and posterior was as follows: precision: 0.47, 0.80, 0.48, 0.68, 1, and 0.85, respectively; recall: 0.54, 0.97, 0.59, 0.78, 0.81, and 0.44 (training dataset), respectively; precision: 0.62, 0.76, 0.57, 0.75, 0.98, and 0.93, respectively; and recall: 0.74, 0.99, 0.66, 0.72, 0.76, and 0.57 (test dataset), respectively. The average accuracy was 0.72 and 0.67, the average precision was 0.71 and 0.77, the average recall was 0.69 and 0.74, and the F1 score was 0.70 and 0.75 for the training and test datasets, respectively (Figure 3).

## 4. Discussion

This study showed that the average accuracy of the machine learning for the training and test data sets was 0.72 and 0.68, respectively. The corresponding average precision was 0.71 and 0.77, respectively, and the average recall was 0.69 and 0.74, respectively.

Many people experience lumbar herniated discs [21,22]. Physical examination, MRI, and electromyography are approaches commonly used to diagnose this disease. However, there are limitations to diagnosis through physical examination [2]; MRI also has limited diagnostic performance and is more expensive than CT [3]. Electromyography is also limited in terms of its invasiveness, causing pain to patients. Hence, the authors used DITI as an alternative diagnostic approach. To the best of our knowledge, this is the first study to apply machine learning to DITI for performing differential screening and classifying L5 radiculopathy and S1 radiculopathy.

In previous reports, research on infrared thermography has been actively conducted in other diseases. DITI can objectively visualize changes in skin temperature in patients, when it is performed with the validated method [23,24]. On the other hand, Vreugdenburg et al., reported that DITI is not yet validated in breast examination, although DITI is still being studied [25], and Mambou et al., reported that DITI can be applicable for breast cancer detection using a deep learning model [26]. Magalhaes et al., showed that infrared thermography can also be used for skin cancer screening and monitoring of skin lesions after treatment. However, early-stage melanoma identification is still not achievable [27]. Adam et al., showed that in diabetes patients, diabetic foot disease is an important complication that causes disability and reduces quality of life. Therefore, early detection is important for effective treatment, and various techniques based on infrared thermography will aid in diagnosis; computer-aided diagnosis can also be useful [28]. Park et al., reported that in carpal tunnel syndrome (CTS), DITI uses a different approach depending on the disease duration and severity and may play a complementary role in understanding and evaluating CTS [29]. DITI is also used in sports medicine. Al-Nakhli et al., reported that exercise-induced muscle damage or delayed-onset muscle soreness occurs in athletes who experience pain due to unexpected exercise without prolonged physical activity or who exercise beyond their training limits. It is one of the most common forms of sports injuries. They reported that measurement ambiguity can be resolved with infrared thermography [7]. Infrared thermography was also applied on the spine area. Zhang et al., showed the correlation of back pain with changes in the thermatome [8]. Cho et al., performed statistical analysis and showed that the sensitivity to clinical symptoms was 88.6% and the accuracy of surgical findings was 86.4% in patients with multiple disc lesions, which is of particular clinical value in diagnosing recurrent disc lesions [30]. Ryu et al., used DITI for neurilemmoma of the deep peroneal nerve for the compression test [31].

In Figure 3, the accuracy of the DITI anterior and posterior HLD L4/5 is mainly deteriorated in the process of image discrimination of the developed algorithm. Perhaps this is because the left and right sides are not added as labels. Park et al., reported that the other confusing factors are that DITI shows hyperthermia sometimes on the side of the lesion rather than the thermal pattern of the extremities on the side of the lesion shows hypothermia compared with the opposite, usually when in the acute phase, trauma, and severe pain [32]. Considering those points, if it is applied to the feet of HLD L4/5 and L5/S1 patients, the patient’s history, such as symptom duration, and pain scale, the accuracy will be better. That is, when distinguishing between HLD L4/5 and L5/S1 based on machine learning, it may be technically useful to first determine the DITI of the feet. It would be more accurate if the accuracy was evaluated with two labels on the feet of HLD L4/5 and L5/S1 patients. Considering the real-world DITI acquisition condition, the authors believe that it was meaningful to test the performance of the bag of visual words by performing a multi-classification experiment. As a research topic in the next study, if the related DITI data of one disease label are reduced to two labels using a recurrent neural network and is sequential in an anterior–posterior–feet manner, then the accuracy may further improve.

In this study, there were two limitations. First, the data set is not large enough to train the machine learning model with six labels. Because this study is multi-class machine learning classification, more data will be needed for better accuracy with representatives. Second, balanced accuracy is not so high (approximately 70%) for clinical application. Despite the limitations, the national standard DITI data center has the largest DITI cohort, and now the authors have attempted to label more data with a trained labeler and spine specialist. This study aimed to focus on screening and supporting community-based healthcare, not on specialized diagnosis. Therefore, it would be valuable to obtain those results for a supporting aspect.

Considering screening purpose, authors can analyze the medical cost comparison in the follow-up study. Although the price is slightly different for each hospital, MRI in the lumbar region is about 500 $ and DITI is about 100 $ in the Korean medical field. Authors will further compare the mathematical expenses between MRI and DITI for lumbosacral radiculopathy screening using national healthcare service data.

## 5. Conclusions

In conclusion, the bag of visual words as a machine learning method needs to be improved with the amount of data to overcome the ambiguity of DITI followed by individual time-periodic sympathetic variations and increase the therapeutic effect of primary pain interventions with economical cost. In the future, the authors can improve the accuracy of a multi-class classification and perform the medical cost analysis.

## Figures and Tables

**Figure 1 healthcare-10-01094-f001:**
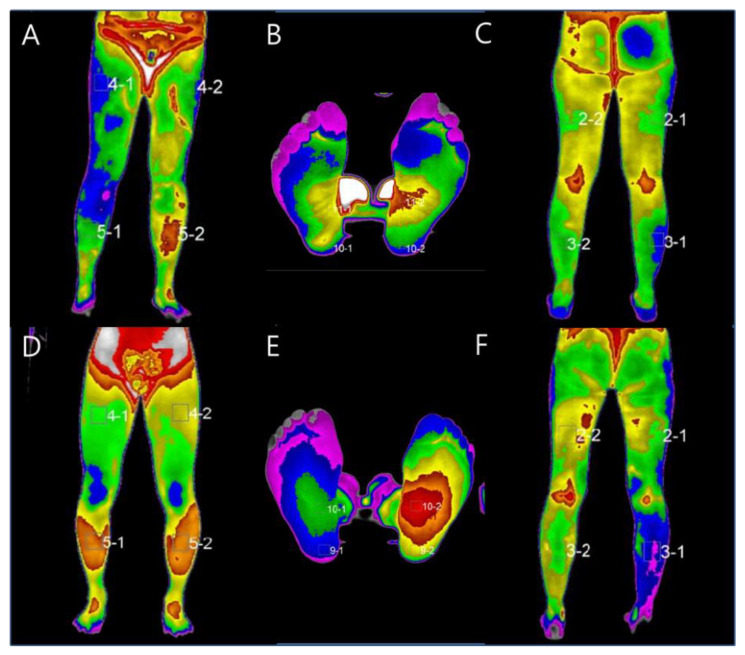
Representative DITI example and labeled sequences. HLD L4/5 patients undergoing DITI anterior (**A**), feet (**B**), and posterior (**C**) were labeled 1, 2, and 3 sequentially, respectively, and HLD L5/S1 patients undergoing DITI anterior (**D**), feet (**E**), and posterior (**F**) were labeled 4, 5, and 6, respectively.

**Figure 2 healthcare-10-01094-f002:**
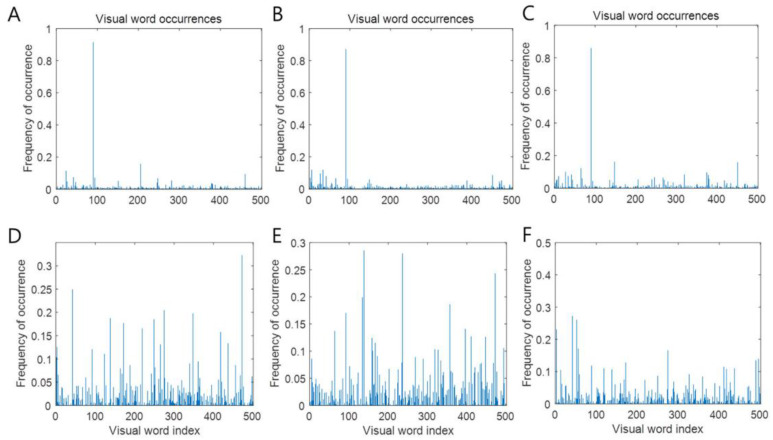
Visual word occurrence of each label. HLD L4/5 patients undergoing DITI anterior (**A**), feet (**B**), and posterior (**C**) and HLD L5/S1 patients undergoing DITI anterior (**D**), feet (**E**), posterior (**F**). Visual word index and frequencies showed characteristics of each image using the bag of visual word image classification algorithm.

**Figure 3 healthcare-10-01094-f003:**
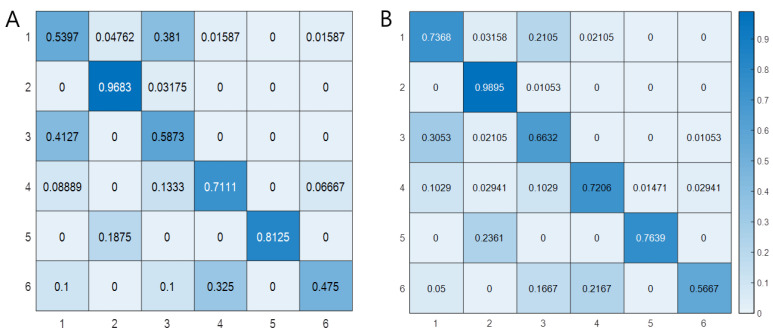
Confusion matrix of the training data set (**A**) and test data set (**B**). The average accuracy of the training set (**A**) is 0.72, and that of the test set (**B**) is 0.68. The corresponding average precision is 0.71 and 0.77, respectively, the average recall is 0.69 and 0.74, respectively, and the F1 score is 0.70 and 0.75, respectively.

**Table 1 healthcare-10-01094-t001:** Age and sex distribution of DITI data.

	HLD L4/5		HLD L5/S1	
Age Group	Male	Female	Male	Female
0–19	2	1	2	0
20–29	18	15	17	10
30–39	33	30	30	26
40–49	55	43	19	27
50–59	49	57	24	28
60–69	62	68	31	45
70	46	55	28	21

HLD, herniated lumbar disc; DITI, digital infrared thermographic image.

## Data Availability

Data can be provided only for legitimate research purposes and upon request to the principal investigator with the anonymization process and the approval of the research ethics committee of the institution.

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
