# Peer review of "Differential Screening of Herniated Lumbar Discs Based on Bag of Visual Words Image Classification Using Digital Infrared Thermographic Images"

_healthcare, 2022, doi:10.3390/healthcare10061094_

Round 1

Reviewer 1 Report

Thank you for allowing me to review this manuscript. This manuscript entitled "Differential Detection of Lumbar Disc Herniations Based on Visual Word Image Classification 2 Bag Using Digital Infrared Thermographic Imaging 3" which aims to perform differential detection of L4-5 and L5-S1 discs, which are of high frequency. herniated lumbar discs (HLD) in patients undergoing radiculopathy because it is expected to help clinicians accurately determine foraminal root block along with diagnosis.

It is an interesting and highly relevant article today, although it has several limitations that make it suitable for publication in this journal. These limitations are detailed below:

- In the introduction, it would be important to point out the relevance and timeliness of the subject of study.

- In the material and methods section it is described in detail. The large number of the sample is a strength of the study. However, it should be reflected if said sample is representative. As a signal and weakness, we can highlight that it should be pointed out in more detail how the data has been collected, as well as the study design.

- In relation to the discussion, the manuscript discusses the results in an orderly manner, with sufficient citations and rigor. However, due to the importance of the subject of study, the number of studies cited in this section will be modified, which will allow justifying the results and comparing them.

- Finally, the conclusions are clear and precise. However, we recommend indicating more precisely the implication in clinical practice of the same. Also, it would be interesting to state the lines of the future that the authors consider.

- In relation to the conclusions, they are clear and precise. However, we recommend pointing out more precisely their implication in clinical practice and the line of future proposed.

- Finally, in relation to bibliographical references, it would be interesting to increase their number. In addition, as stated, the central theme of the manuscript is highly topical, which requires that the bibliography be from recent years. However, in the bibliographical references section, there are articles with more than 10 years. It would be necessary to review this aspect and increase the citation with a more current citation.

nice job

Author Response

- In the introduction, it would be important to point out the relevance and timeliness of the subject of study.

I agree with your opinion. Therefore, we modified the introduction section.

Electromyography is a useful method of diagnosing radiculopathy.[5] However, there is an invasive aspect, which involves inserting a needle into the patient, thereby causing pain during the examination. Otherwise, Digital infrared thermographic imaging (DITI) showed the functional changes of body temperatures noninvasively with economical cost and short examination time.

This study, we aimed to perform differential screening of L4-5 and or L5-S1 discs, which are high-frequency herniated lumbar discs (HLDs),[15,16] and patients from two to five decades have a more than 90 percent chance of HLD occurring either at L4/5 or L5/S1 radiculopathy. [17] Because the dermatome is adjacent and it can be difficult to distinguish it, the authors we try to perform the analysis by applying machine learning such that community medical personnel who are not spine specialists can help in conservative care, such as foraminal block, for patients who were classified based on DITI without using expensive equipment, such as MRI. Therefore, this machine learning algorithm helps clinicians accurately determine the target of foraminal root block with the screening of HLD level. The authors explored the infrared thermographic images of lumbosacral radiculopathy from a multicenter database using machine learning-based infrared thermographic software

- In the material and methods section it is described in detail. The large number of the sample is a strength of the study. However, it should be reflected if said sample is representative. As a signal and weakness, we can highlight that it should be pointed out in more detail how the data has been collected, as well as the study design.

Thank you for your good comments. The authors described more details in the method section and wrote specific attributes used in this study. And add limitation of representatives in the discussion section.

A total of 1282 patients who performed 1 level discectomy from 2006 to 2020 were retrospectively reviewed by specially trained spine surgeons, and 842 patients were enrolled in this study. The use of patient data retrospectively for research purposes was approved, and informed consent was waived by the Institutional Review Board at National Health Insurance Service Ilsan Hospital (IRB no.: 2021-07-030). The dataset (National Reference Standard Korean Thermal Data Center at National Health Insurance Service Ilsan Hospital and Gangnam Severance Hospital) of patients who underwent one-level discectomy was applied to develop and to externally validate an machine learning-based classification tool for thermographic images.

In this study, there were two limitations. First, the data set is not large enough to train the machine learning model with six labels. Because this study is multi-class machine learning classification, more data will be needed for better accuracy with representatives. Second, balanced accuracy is not so high (approximately 70%) for clinical application.

- In relation to the discussion, the manuscript discusses the results in an orderly manner, with sufficient citations and rigor. However, due to the importance of the subject of study, the number of studies cited in this section will be modified, which will allow justifying the results and comparing them.

I agree with your excellent comment. Therefore, add more references for justifying the results and comparing them.

In previous reports, research on infrared thermography has been actively conducted in other diseases. DITI can objectively visualize changes in skin temperature in patients, when it performed with validated method.[28, 29] On the other hands, Vreugdenburg et al. reported that DITI is not yet validated in breast examination, although DITI is still being studied.[20]

...

In Figure 3, the accuracy of the DITI anterior and posterior HLD L4/5 is mainly deteriorated in the process of image discrimination of the developed algorithm. Perhaps it is because the left and right sides are not added as labels. Park et al. reported that other confusing factors are that DITI shows hyperthermia sometimes on the side of the lesion rather than the thermal pattern of the extremities of the side of the lesion shows hypothermia compared to the opposite in usual when acute phase, trauma, and severe pain [31]. Considering those points, if it is applied to the feet of HLD L4/5 and L5/S1 patients, patient's history such as symptom duration, and pain scale, the accuracy will be better.

- the conclusions are clear and precise. However, we recommend indicating more precisely the implication in clinical practice of the same. Also, it would be interesting to state the lines of the future that the authors consider.  - In relation to the conclusions, they are clear and precise. However, we recommend pointing out more precisely their implication in clinical practice and the line of future proposed.

I agree with your good comment. Therefore, modify the conclusion as follows

In conclusion, bag of visual words as a machine learning method need to be improved with the amount of data to overcome the ambiguity of DITI followed by individual time-periodic sympathetic variations and increase the therapeutic effect of primary pain interventions with economical cost. In the future, the authors can improve the accuracy of a multi-class classification and perform the medical cost analysis.

- Finally, in relation to bibliographical references, it would be interesting to increase their number. In addition, as stated, the central theme of the manuscript is highly topical, which requires that the bibliography be from recent years. However, in the bibliographical references section, there are articles with more than 10 years. It would be necessary to review this aspect and increase the citation with a more current citation.

I agree with your appropriate comment. Therefore, add a more current citations as follows in the bibliographical and discussion section.

Ammer, K., & Ring, F. (2019). The thermal human body: a practical guide to thermal imaging. Jenny Stanford Publishing

Park, T. Y., Son, S., Lim, T. G., & Jeong, T. (2020). Hyperthermia associated with spinal radiculopathy as determined by digital infrared thermographic imaging. Medicine, 99(11).

Mambou, S. J., Maresova, P., Krejcar, O., Selamat, A., & Kuca, K. (2018). Breast cancer detection using infrared thermal imaging and a deep learning model. Sensors, 18(9), 2799.

In previous reports, research on infrared thermography has been actively conducted in other diseases. DITI can objectively visualize changes in skin temperature in patients, when it is performed with a validated method.[Ammer, K., & Ring, F. (2019)] On the other hand, Vreugdenburg et al. reported that DITI is not yet validated in breast examination, although DITI is still being studied,[20] and Mambou et al. reported that DITI can be applicable for breast cancer detection using deep learning model[21].

In Figure 3, the accuracy of the DITI anterior and posterior HLD L4/5 has mainly deteriorated in the process of image discrimination of the developed algorithm. Perhaps it is because the left and right sides are not added as labels. Park et al. reported that other confusing factors are that DITI shows hyperthermia sometimes on the side of the lesion rather than the thermal pattern of the extremities of the side of the lesion shows hypothermia compared to the opposite in usual when acute phase, trauma, and severe pain [31]. Considering those points, if it is applied to the feet of HLD L4/5 and L5/S1 patients, patient's history such as symptom duration, and pain scale, the accuracy will be better

Reviewer 2 Report

In Greece it is extremely uncommon to use DITI as a diagnostic approach on lumbar disc herniation - radiculopathy. We mostly rely on the  MRI and physical examination findings. Thus, I do not consider myself an expert on judging the diagnostic approach with DITI. Nevertheless, the article was very interesting and intriguing to me. It would, also, have been interesting if the authors performed a cost analysis or a comparison of costs between MRI-DITI, but let this be an extra incentive for future research for them.

Author Response

Thank you for your great comments.

Considering screening purposes, authors can analyze the cost of a follow-up study. Although the price is slightly different for each hospital, MRI in the lumbar region is about 500 $ and DITI is about 100$ in the Korean medical field. Using national healthcare service data, the authors will further compare the mathematical expenses between MRI and DITI for lumbosacral radiculopathy screening.

Reviewer 3 Report

The manuscript presents a research that aims to perform differential screening of L5 and S1 lumbosacral radiculopathy using digital infrared thermal imaging and machine learning algorithms.

The proposed method achieved an average accuracy of 0.72 and 0.67, the average precision of 0.71 and 0.77, the average recall of 0.69 and 0.74, and a F1 score of 0.70 and 0.75 for the training and test datasets.

I find the topic interesting and being worth of investigation and the document is well strucutred, organized, fluidly written, the background is adequate, the methodology well explained, results are clearly presented, supporting the discussion and conclusions.

Although I propose the following comments/suggestions:

- The abstract is poorly descriptive of the content, it should be better organized: problem, motivation, aim, methodology, main results, further impact of those results.

- keywords should be in alphabetical order.

- I strongly suggest authors from refraining using personal pronouns such as "we" and "our" throughout the text and I encourage them to write it in an impersonal form of writing.

- The authors do not specify the atributes used in the methods, for example the initial guess of the k-means clusterings and the SVM kernel.

- The study limitations are not present at discussion section.

- No further research is proposed at conclusions.

- Important references are missing:

Ring, E. F. J., & Ammer, K. (2012). Infrared thermal imaging in medicine. Physiological measurement, 33(3), R33.

Ammer, K., & Ring, F. (2019). The thermal human body: a practical guide to thermal imaging. Jenny Stanford Publishing.

Author Response

- The abstract is poorly descriptive of the content, it should be better organized: problem, motivation, aim, methodology, main results, further impact of those results.

I agree with your comments. We revised abstract as follow

Doctors in primary hospitals can get the impression of lumbosacral radiculopathy with a physical exam and need to acquire medical images such as an expensive MRI for diagnosis. Then, doctors will perform a foraminal root block to the target root for pain control. However, there was insufficient screening medical image examination for precise L5 and S1 lumbosacral radiculopathy, which is most prevalent in the clinical field. Therefore, to perform differential screening of L5 and S1 lumbosacral radiculopathy, we applied digital infrared thermographic images (DITI) to the machine learning (ML) algorithm, which is the bag of visual words method. Our dataset included data from the healthy population and radiculopathy patients with herniated lumbar discs (HLDs) L4/5 and L5/S1. A total of 842 patients were enrolled and split the dataset into a 7:3 ratio as the training algorithm and test dataset to evaluate model performance. The average accuracy was 0.72 and 0.67, the average precision was 0.71 and 0.77, the average recall was 0.69 and 0.74, and the F1 score was 0.70 and 0.75 for the training and test datasets. Application of bag of visual words algorithm to DITI classification will aid in the differential screening of lumbosacral radiculopathy and increase the therapeutic effect of primary pain interventions with economical cost.

- keywords should be in alphabetical order.

Thank you for your good comment. Keywords were changed as follows in alphabetical order

bag of visual words; infrared thermography; lumbosacral radiculopathy; machine learning

- I strongly suggest authors from refraining using personal pronouns such as "we" and "our" throughout the text and I encourage them to write it in an impersonal form of writing.

Thank you for your excellent comment. The manuscript was changed with impersonal form as much as possible.

- The authors do not specify the atributes used in the methods, for example the initial guess of the k-means clusterings and the SVM kernel.

Thank you for your good comment. We described the specific attributes of k-means clustering and kernel of SVM in the method section.

k-means clustering(initial guess of 6928800 points and 500 centroid), codebook generation, and learning and recognition (making the feature vector and support vector machine(radial basis function kernel) classify an image set

- The study limitations are not present at discussion section.

Thank you for your good comment.

We describe limitations at discussion section with more details

#In this study, there were two limitations. First, the data set is insufficient to train the machine learning model with six labels. Because this study is a multi-class machine learning classification, more data will be needed for better accuracy. Second, balanced accuracy is not so high (approximately 70%) for clinical application.

- No further research is proposed at conclusions.

Thank you for your valuable comment.

We add further research plans in discussion and conclusion section.

#Considering screening purposes, authors can analyze the cost of a follow-up study. Although the price is slightly different for each hospital, MRI in the lumbar region is about 500 $ and DITI is about 100$ in the Korean medical field. Using national healthcare service data, the authors will further compare the mathematical expenses between MRI and DITI for lumbosacral radiculopathy screening.

#In the future, we can improve the amount of data and accuracy of classification and perform the medical cost analysis.

- Important references are missing:

  1. Ring, E. F. J., & Ammer, K. (2012). Infrared thermal imaging in medicine. Physiological measurement, 33(3), R33.
  2. Ammer, K., & Ring, F. (2019). The thermal human body: a practical guide to thermal imaging. Jenny Stanford Publishing

Thank you for your recommendation with good references. We carefully investigated recommended references and add the citations in the introduction section.

Digital infrared thermographic imaging (DITI) can objectively visualize changes in skin temperature in patients [7, 8], especially those undergoing radiculopathy. [9]

Reviewer 4 Report

1. The manuscript is overall well written

2. The idea is novel. However, it is not something commonly used. Therefore, this technology needs to be better explained in the methods section. Otherwise, a majority of the manuscript is difficult to follow; and the purpose hard to understand

3. Pls discuss the literature on this technology more elaborately. The discussion is very short. Pls add the reference. Park, Tae Yoon MDa; Son, Seong MD, PhDa,; Lim, Tae Gyu MDb; Jeong, Taeseok MD, PhDa Hyperthermia associated with spinal radiculopathy as determined by digital infrared thermographic imaging, Medicine: March 2020 - Volume 99 - Issue 11 - p e19483 doi: 10.1097/MD.0000000000019483

4. What is the clinical purpose or application of the concerned test? Where does it stand in comparison with the existing technology? What was the background of using this technology for evaluating lumbar radiculopathy? Pls elaborate

5. While the purpose of DITI in lumbar radiculopathy itself is not well-validated, what is the purpose of incorporating machine learning to it? The authors need to explain this clearly

6. Why only specific levels L45 and L5S1 were included? Pls elaborate

7. The conclusion section needs to be totally modified. The accuracy and precision of this technology is still not good. However, the conclusion seems very optimistic.

8. The authors needs to elaborate the future of this technology

Author Response

The authors appreciate your excellent comments.

  1. The idea is novel. However, it is not something commonly used. Therefore, this technology needs to be better explained in the methods section. Otherwise, a majority of the manuscript is difficult to follow; and the purpose hard to understand

Thank you for your excellent comments.

The authors add some references to digital infrared thermographic images in the introduction section and describe the initial parameters of the k means clustering method and specific Kerner used for SVM in the method section. The purpose of the study and machine learning algorithm was modified in the introduction section.

Digital infrared thermographic imaging (DITI) can objectively visualize changes in skin temperature in patients [7, 8], especially those undergoing radiculopathy.[9]

This study aimed to perform differential screening of L4-5 or L5-S1 discs, which are high-frequency herniated lumbar discs (HLDs),[15,16] and patients from two to five decades have a more than 90 percent chance of HLD occurring either at L4/5 or L5/S1 radiculopathy. [17] 

The steps are feature extraction, which selects feature point locations using the grid method and extracting speeded up robust features from the selected feature point locations, k-means clustering(initial guess of 6928800 points and 500 centroids), codebook generation, and learning and recognition (making the feature vector and support vector machine(radial basis function kernel) classify an image set).[17]

  1. Pls, discuss the literature on this technology more elaborately. The discussion is concise. Pls, add the reference. Park, Tae Yoon MDa; Son, Seong MD, PhDa,; Lim, Tae Gyu MDb; Jeong, Taeseok MD, PhDaHyperthermia associated with spinal radiculopathy as determined by digital infrared thermographic imaging, Medicine: March 2020 - Volume 99 - Issue 11 - p e19483 DOI: 10.1097/MD.0000000000019483

Thank you for your recommendation with good references. We carefully investigated recommended references and added the citations in the discussion section.

In Figure 3, the accuracy of the DITI anterior and posterior HLD L4/5 has mainly deteriorated in the developed algorithm's image discrimination. Perhaps it is because the left and right sides are not added as labels. Park et al. reported that other confusing factors are that DITI shows hyperthermia sometimes on the side of the lesion rather than the thermal pattern of the extremities of the side of the lesion shows hypothermia compared to the opposite in usual when acute phase, trauma, and severe pain [31]. Considering those points, if it is applied to the feet of HLD L4/5 and L5/S1 patients, patient's history such as symptom duration, and pain scale, the accuracy will be better.

  1. What is the clinical purpose or application of the concerned test? Where does it stand in comparison with the existing technology? What was the background for using this technology to evaluate lumbar radiculopathy? Pls elaborate

Thank you for your good question. Answers are as follows,

described and modified in the introduction sections

Digital infrared thermographic imaging (DITI) can objectively visualize changes in skin temperature in patients [7, 8], especially those undergoing radiculopathy.[9]

Community medical personnel who are not spine specialists can help in conservative care, such as foraminal block, for patients classified based on DITI without using expensive equipment, such as MRI. Therefore, this machine learning algorithm helps clinicians accurately determine the target of foraminal root block with the screening of HLD level.

  1. While the purpose of DITI in lumbar radiculopathy itself is not well-validated, what is the goal of incorporating machine learning into it? The authors need to explain this clearly.

Thank you for your excellent comment. We modified and summarized it in the conclusion section.

The bag of visual words as a machine learning method needs to be improved with the amount of data to overcome the ambiguity of DITI followed by individual time-periodic sympathetic variations and increase the therapeutic effect of primary pain interventions with economical cost.

  1. Why only specific levels L45 and L5S1 were included? Pls elaborate

Thank you for a good question. Because specific levels L45 and L5S1 are the most prevalent in the clinical field. The authors explain the reason and references for specific levels in the introduction section.

This study aimed to perform differential screening of L4-5 or L5-S1 discs, which are high-frequency herniated lumbar discs (HLDs),[15,16] and patients from two to five decades have a more than 90 percent chance of HLD occurring either at L4/5 or L5/S1 radiculopathy. [17] 

Therefore, this machine learning algorithm helps clinicians accurately determine the foraminal root block with the screening of HLD level.

  1. The conclusion section needs to be totally modified. The accuracy and precision of this technology is still not good. However, the conclusion seems very optimistic.

I agree with you. Conclusion sections were modified as follows

In conclusion, the bag of visual words as a machine learning method needs to be improved with the amount of data to overcome the ambiguity of DITI followed by individual time-periodic sympathetic variations and

  1. The authors needs to elaborate the future of this technology

Thank you for your valuable comments

Applying this algorithm to DITI classification will aid in the differential screening of lumbosacral radiculopathy and increase the therapeutic effect of primary pain interventions at an economical cost. Therefore, we modified the conclusion section as follows.

In conclusion, the bag of visual words as a machine learning method needs to be improved with the amount of data to overcome the ambiguity of DITI followed by individual time-periodic sympathetic variations and increase the therapeutic effect of primary pain interventions with economical cost. In the future, we can improve the accuracy of a multi-class classification and perform the medical cost analysis.

Round 2

Reviewer 4 Report

The recommended changes have been incorporated. We congratulate the authors.

This manuscript is a resubmission of an earlier submission. The following is a list of the peer review reports and author responses from that submission.